# Disease Perception and Mental Health in Pregnancies with Gestational Diabetes—PsychDiab Pilot Study

**DOI:** 10.3390/jcm12103358

**Published:** 2023-05-09

**Authors:** Claudia Rieß, Yvonne Heimann, Ekkehard Schleußner, Tanja Groten, Friederike Weschenfelder

**Affiliations:** Department of Obstetrics, University Hospital Jena, Friedrich-Schiller-University, 07747 Jena, Germany

**Keywords:** gestational diabetes, mental health, wellbeing, SCL-90-R, pregnancy

## Abstract

(1) Background: The aim of this work is to investigate the extent to which pregnant women’s well-being is burdened by the diagnosis of gestational diabetes, as well as their sensitivities and illness perceptions. Since gestational diabetes is associated with mental disorders, we hypothesized that the burden of illness might be related to pre-existing mental distress. (2) Methods: Patients treated for gestational diabetes in our outpatient clinic were retrospectively asked to complete a survey, including the self-designed Psych-Diab-Questionnaire to assess treatment satisfaction, perceived limitations in daily life and the SCL-R-90 questionnaire to assess psychological distress. The association between mental distress and well-being during treatment was analyzed. (3) Results: Of 257 patients invited to participate in the postal survey, 77 (30%) responded. Mental distress was found in 13% (n = 10) without showing other relevant baseline characteristics. Patients with abnormal SCL-R-90 scores showed higher levels of disease burden, were concerned about glucose levels as well as their child’s health, and felt less comfortable during pregnancy. (4) Conclusions: Analogous to the postpartum depression screening, screening for mental health problems during pregnancy should be considered to target psychologically distressed patients. Our Psych-Diab-Questionnaire has been shown to be suitable to assess illness perception and well-being.

## 1. Introduction

Recent studies show a prevalence of nearly 10% of gestational diabetes mellitus (GDM) in Germany [1] which means over 70,000 pregnant women per year are confronted with the diagnosis of GDM. Interviews conducted by Evans et al. revealed various factors that burdened the patients during their pregnancy, including the restriction of autonomy, permanent external as well as internal monitoring, instruction and negative reactions by medical professionals and uncertainty about the influence of GDM on their own future as well as that of the child [2]. Depression tendencies, anxiety, and stress have been shown to be more pronounced in GDM patients at the time of diagnosis and can change over the course of pregnancy [3,4]. Studies on the quality of life in pregnancies with GDM as well as possible influencing factors vary greatly in their statements, whereby multiple factors such as ’maternal body mass index (BMI), mode of delivery and socioeconomic factors seem to have a major impact [5,6,7]. Regarding patients’ satisfaction with therapeutic interventions, Mautner and Dorfer found that treatment with insulin has no negative effects on the emotional state of pregnant women [3], while many patients reported concerns about insulin therapy in a study by Draffin et al. [4]. Furthermore, Hinkle et al. illustrate the already-known mutual association between depression and GDM in a longitudinal study. They demonstrate that pregnant women with depression scores in the upper quartile in the first two trimesters of their pregnancy had a significantly increased risk of GDM after adjustment for covariates, e.g., prepregnancy BMI, age and ethnicity with depressive symptoms correlating linearly with the risk of GDM. Additionally, it was shown by multiple studies that GDM increased the risk of postpartum depression by up to 4.6 times [8].

The aim of this work is to investigate the extent to which the well-being of pregnant women in our study cohort was burdened by the diagnosis of GDM as well as their sensitivities and illness perceptions during pregnancy. We hypothesized that mental distress and personality accentuation, as retrospectively assessed by the SCL-R-90 questionnaire, would influence treatment satisfaction during GDM treatment and thus overall outcome.

## 2. Materials and Methods

### 2.1. Study Population

The primary cohort of this study consists of 291 singleton pregnancies of patients diagnosed with and treated for GDM in the Competence Center for Diabetes and Pregnancy at our tertiary care hospital from 1 January 2017 until 31 December 2018. The diagnosis of GDM was according to IADPSG and WHO-2013 criteria [9,10]. Diabetes care was provided according to the German S3 guidelines published in 2018 and provided by our hospital-based outpatient department by a specialized team of obstetricians, diabetologists, diabetes consultants, midwives and nurses [11]. The cohort was monitored every four weeks in the case of diet control and fortnightly in insulin-treated women. Ethical approval was given by the local Ethical Committee of the Friedrich-Schiller-University, Jena, Germany (2019/1557-Bef).

### 2.2. Study Questionaire

We developed a questionnaire regarding treatment and therapy as well as treatment satisfaction. All included patients received this questionnaire either 6 to 12 weeks after delivery as part of postnatal GDM care or, because of the retrospective study design, no later than one year after the index pregnancy. The Psych-Diab-Questionnaire consist of 25 questions about their adherence to and perception of treatment (e.g., dietary changes, eliminating sweets and/or regular exercise), what they paid attention to after delivery (e.g., oral glucose tolerance test after 6 to 12 weeks, maintaining dietary changes and/or long breastfeeding) and what measures they found helpful. Other questions relate to well-being during pregnancy, e.g., whether patients were concerned about their own health or that of their child, whether they felt restricted in their daily lives and the effects of exercise and dietary changes. We used an ordinal scale from 0 to 10 with 0 being the lowest and 10 being the highest agreement to the questions asked. 

Additionally, the Symptom Checklist-90-R (SCL-90-R), a self-report questionnaire to evaluate different psychological problems and symptoms [12] was part of the survey. The SCL-90-R is a 90-item self-report instrument that helps to evaluate a broad range of psychological problems and symptoms of psychopathology across nine subscales (somatization, obsessive-compulsive, interpersonal sensitivity, depression, anxiety, hostility, phobic anxiety, paranoid ideation, and psychoticism) and three global scales (Global Severity Index, Positive Symptom Distress Index, and Positive Symptom Total). Participants rate the severity of their symptoms on a scale of 0–4 (0 = not at all, 1 = a little bit, 2 = moderate, 3 = strong, or 4 = very strong). According to Derogatis, a case is defined with GSI ≥ 63 or at least two of the primary scales with T-scores ≥ 63. This case definition was also used to categorize our patients into subgroups: women with mental distress and women without mental distress. T-Scores over 60 in the global indices indicate overall mental distress, whereas the T-scores of the primary scales are organized as ‘mildly to noticeably elevated’ with T-scores from 60 to 69 and ‘severely to very severely elevated’ with T-scores from 70 to 80 [12].

### 2.3. Study Data Collection

We collected basic characteristics as well as patients’ and family medical history from hospital records if patients gave their consent to use the data for research. Calculated items included gestational weight gain classified after IOM and calculated by difference of prepregnancy weight and last documented weight during pregnancy [13], BMI from maternal height and prepregnancy weight according to WHO [14] and change of HbA1c using the first retrieved HbA1c level according to IFCC or NGSP/DCCT standard and the last one before delivery while obtained every four weeks. Perinatal outcome data included fetal birth and birth weight percentiles according to Voigt et al. [15]. Neonates were grouped into LGA (large for gestational age; fetal growth above 90th percentile) and SGA (small for gestational age; fetal growth below the 10th percentile) according to birth weight, gestational age and sex. Further neonatal outcome data were 5-min Apgar score, postnatal admission to neonatal intensive care unit (NICU), neonatal hypoglycemia and hyperbilirubinemia. Outcome data were retrieved from the standardized nationwide used perinatal documentation systems of our university hospital and patients’ maternity records.

### 2.4. Statistical Analysis

Statistical analysis was performed with SPSS 27.0 (IBM Corp. Released 2021. IBM SPSS Statistics for Windows, Version 27.0. IBM Corp, Armonk, NY, USA). The Chi2 test or Fisher exact test was used to compare categorical data. We used median and interquartile ranges for data presentation as most continuous data were not normally distributed. Non-parametric tests were performed to compare continuous data between subgroups ‘no mental distress’ vs. ‘mental distress’, defined by the case definition of Derogatis (a case is defined with GSI ≥ 63 or at least two of the primary scales with T-scores ≥ 63) [12]. A *p*-value < 0.05 was considered to indicate statistical significance (2-tailed).

## 3. Results

Of 291 women treated with GDM in our outpatient clinic in 2017 and 2018, 257 could be contacted and received the study questionnaire; 77 (30%) questionnaires were returned and were eligible for statistical analysis. (See Figure 1) We did not see relevant differences between women responding and not responding concerning major baseline and perinatal outcome characteristics (see Appendix A Table A1).

### 3.1. Baseline Characteristics

Table 1 shows baseline characteristics of the total cohort and the subgroups with and without mental distress as evaluated with SCL-R-90 at the time of GDM diagnosis. Psychological abnormalities in the form of mental distress were found in 13% of (n = 10) women. Maternal characteristics in women with abnormal SCL-R-90 scores did not differ from those with normal scores, except in BMI (29 kg/m^2^ vs. 25 kg/m^2^). No differences were found concerning maternal age, blood pressure, family history of diabetes, history of diagnosed psychiatric disorders, social parameters, fetal ultrasound parameters or results of 75 g oGTT. T-scores of all subscales and global scales differed significantly between the two subgroups with and without mental distress (see Table 1).

### 3.2. Pregnancy and Perinatal Outcome

Table 2 shows pregnancy characteristics and perinatal outcomes. Data on perinatal outcome could only be evaluated from 61 cases due to the retrospective design: 55 in the No Mental Distress group and six cases in the mental distress group. There was no difference concerning treatment methods, pregnancy complications, gestational weight gain and necessity of induction of labor, rate of cesarean section, birth weight, hyperbilirubinemia or admission to NICU. Gestational age at delivery differed significantly (39 weeks vs. 38 weeks, *p* = 0.031) but without clinical relevance.

### 3.3. Association of Psychological Distress and Overall Well-Being during Pregnancy Obtained by the Psych-Diab-Questionnaire

Table 3 shows the results of the Psych-Diab-Questionnaire for the total cohort and the subgroups. Patients with mental distress felt significantly less comfortable during pregnancy (7 vs. 4.5). They also feared bad blood sugar values more, not only while measuring (6 vs. 9) but also while eating (7 vs. 10) and had more worries that the high blood sugar levels would put their child’s health at risk (8 vs. 10). The diagnosis made patients with mental distress worry more about their child’s health (7 vs. 10) as well as making them feel more burdened (6 vs. 8). In addition, they were less likely to be positively surprised by the reactions in their environment after diagnosis (5 vs. 0). There were no significant differences regarding the questions about feeling restricted in everyday life by therapy measures, but patients without mental distress had more problems with their change in diet (7 vs. 3.5) (see Figure 2).

## 4. Discussion

In our study cohort, 13% were revealed to have abnormal SCL-R-90 scores revealing mental distress. In the Psych-Diab-Questionnaire, these patients showed higher rating values in the questions asking for disease burden and anxiety and lower rating values in questions evaluating well-being and positive pregnancy perceptions. Patients with abnormal SCL-R-90 scores showed significantly higher levels of disease burden, anxiety about the health of their child, high glucose values when measuring blood glucose or eating, and high blood glucose affecting the health of their child. They were less often positively surprised by the reactions in their environment to the diagnosis of GDM, felt less comfortable during pregnancy and described the change in diet less often as being easy for them. (See Figure 2) Thus, this study provides evidence that mental health is an important factor in the management of GDM that still is underestimated and should be implemented into our daily practice and guidelines. Our findings strongly suggest that patients with psychological distress need more intensive support and that screening for mental impairments at the time of diagnosis of GDM should be considered to identify those patients.

For baseline characteristics, patients with abnormal SCL-R-90 scores had significantly higher values in prepregnancy BMI. In recently published studies, the results regarding the association of psychological abnormalities and BMI are inconsistent. While Hayashi et al. could not find significant differences for depressive symptoms and maternal weight [16], Danyliv et al. published a study where health-related quality of life was impaired by increased BMI and abnormal glucose tolerance after delivery [7]. In a qualitative study by Jarvie et al., it was shown through sequential in-depth interviews that women with a BMI above 30 kg/m^2^ felt stigmatized by healthcare providers and perceived the needed lifestyle changes as unrealistic and counterproductive, especially when they were of low socioeconomic status [17]. This may be a reason why the subgroup with mental distress found it harder to implement lifestyle changes in our survey (7 vs. 3.5). This topic requires further in-depth consideration, as lifestyle and dietary change are one of the core elements of a successful treatment regime.

In our study, pregnancy and neonatal outcome showed no differences between the subgroups. Therefore, it can be assumed that possible influencing factors, such as the need for a cesarean section or admission to the NICU, have not influenced the perceived satisfaction and well-being during pregnancy albeit the retrospective design of this study.

When looking at the findings of our Psych-Diab-Questionnaire (see Table 3), there was a significant difference between the two groups in eight of the 25 items. While patients with higher SCL-R-90 scores perceived the diagnosis of GDM significantly more as a burden (6 vs. 8), the subgroup without mental distress also considered the diagnosis a moderate burden during their pregnancy, but in contrast to the mentally distressed subgroup, they nevertheless felt quite comfortable during pregnancy (7 vs. 4.5). This view of GDM as a burden has recently gained more attention, for example, as highlighted by Craig et al.. They found that the added responsibility as well as financial problems and conflicts with cultural practices impaired the women’s well-being during pregnancy [18]. In addition, Muhwava et al. could show in in-depth interviews with GDM patients that the current biomedical model used for managing GDM fails to implement mental health support during pregnancy. The recommendations by the authors include routine mental health screenings throughout pregnancy as well as post-partum [19]. While screening with the Edinburgh Postnatal Depression Scale (EPDS) is suggested around the same time as the postpartum oGTT in the German guideline [11], screening during pregnancy is not implemented in the guideline, although there are several studies highlighting the association of GDM and depression not only after, but also before delivery. Recently, Hayashi et al. could show heightened CES-D scores in women diagnosed with GDM [16], which was also found by Lee et al., who showed a high prevalence of anxious and depressive symptoms among patients with GDM [20]. Additionally, Pace et al. found a nearly twofold increased risk of being diagnosed with depression during pregnancy in women with GDM [21]. Those findings are consistent with the LINDA-Brazil study by Damè et al., who found depressive symptoms during pregnancy in about 31% of study participants [22]. The association of prepartum depressive symptoms shown by those studies is in line with our results and highlights the importance of mental health screenings during pregnancies with GDM. Analogous to the postpartum depression screening, screening for psychological problems already during pregnancy should be considered in order to be able to specifically care for psychologically burdened patients.

Our Psych-Diab-Questionnaire also showed that women with mental distress were significantly less likely to be positively surprised by the reactions in their environment to the diagnosis. Various studies emphasize the importance of a functioning and healthy social environment, for example, as Kragelund et al. found. They identified several barriers for patients but could show that those barriers could be attenuated by social support as well as high-quality interactions with healthcare providers [23]. Similarly, Gilbert et al. highlighted in a systematic review that a secure social system acted as an important enabler for physical activity and dietary choice [24], as well as Ansarzadeh et al., who showed that social security had the highest effect on the quality of life of patients with GDM [25]. Those results are in accordance with the findings of a systematic review of qualitative studies by Faal Siahkal et al., who found that treatment individualized to the psychosocial needs of the specific patient could improve the management of GDM [26].

While there was no significant difference in our results for feeling restricted by treatment measures such as measuring blood glucose, dietary change and injecting insulin, those therapy interventions made both subgroups feel overall restricted. Especially the women with insulin-dependent GDM experienced this as a strain with a median of 9 in the total cohort. This impairment of quality of life could also be shown by Lee et al. [27] and Figueroa Gray et al. [28], whereas Pantzartzis could not find a difference in quality of life for dietary or insulin-based therapy regimes [29]. Although our data did not find a difference between mental distress for patients with and without insulin therapy, this group of patients may need special care in terms of their mental health. In this context, in a recently published study, Gilbert et al. found that in addition to the metabolic outcome and stress perception, well-being was improved when physical activity was included in the therapy alongside dietary measures, which could counteract the risk of postpartum depression [24]. Physical activity was perceived as beneficial by many patients but was difficult to incorporate into daily life [4]. In a study by Martis et al., suggestions to increase adherence to treatment and thus outcomes were given, including group training, recommendations adapted to the cultural background, as well as training, diaries and consultations in the respective mother tongue [30]. On the other hand, Draffin et al. showed that many patients found it difficult to accept the recommended lifestyle changes. Different eating habits were problematic to implement due to small portions, difficulty in resisting temptations and low variance in meals. Optimized and individualized GDM therapy shows influence on maternal metabolic control as well as psychological well-being and neonatal outcome, as it was shown that patients with more positive attitudes and higher treatment satisfaction had numerically better glycemic control [31]. Additionally noteworthy is the high reinsuring value retrieved for the scheduled ultrasound checks (Figure 2). These data suggest that disease perception and treatment success could be optimized when individual management strategies are applied and patients at risk are identified.

### Strengths and Limitations

A strength of our study is the comparison of a large number of GDM cases treated at one single unit following the same treatment standards and guidelines. Nevertheless, there are some limitations of the presented study that might include the retrospective design of the survey and the consequential time difference between GDM pregnancy and the survey (ranging from 6 to 12 weeks postpartum to one year), especially since the SCL-R-90 questionnaire was used to assess the mental status of the patients is only validated for the psychological symptoms in the last seven days. Additionally, due to the retrospective design, patients with impaired mental health might be underrepresented, since this group is less likely to respond to a survey. The Psych-Diab-Questionnaire used for this study was used for the first time and results might be considered as a preliminary evaluation since it is not yet validated. However, the discriminative results retrieved by the questionnaire underscore its value. For a better classification of the data, a comparison with a healthy control group would have been helpful as well as the longitudinal evaluation of changes in mental health during pregnancy.

## 5. Conclusions

We found a high level of treatment satisfaction in our cohort where high levels of mental distress markedly affected well-being during pregnancy in GDM patients. This study provides evidence that mental health is an important factor in the management of GDM that still is underestimated and should be implemented into daily practice. In order to optimize treatment strategies in GDM patients and to release the burden of disease, care providers need to record the psychological distress of their patients at the time of diagnosis of GDM. Our findings strongly suggest that patients with psychological distress need more intensive support and that screening at the time of diagnosis of GDM should be considered to determine those patients and to be included in GDM guidelines. The Psych-Diab-Questionnaire was revealed to be suitable to evaluate disease perception and well-being in GDM patients.

## Figures and Tables

**Figure 1 jcm-12-03358-f001:**
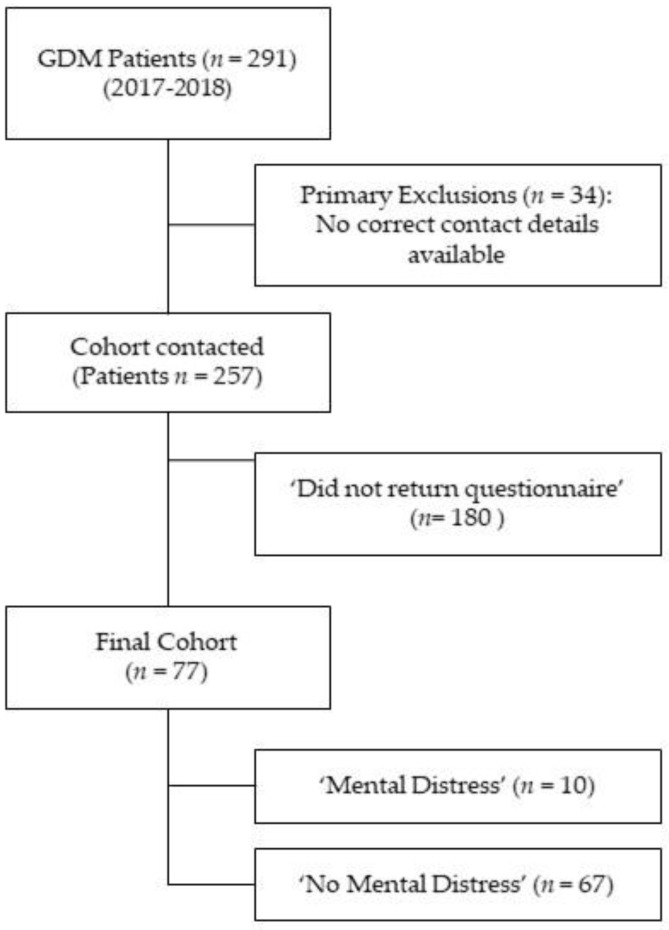
Flowchart of the study cohort.

**Figure 2 jcm-12-03358-f002:**
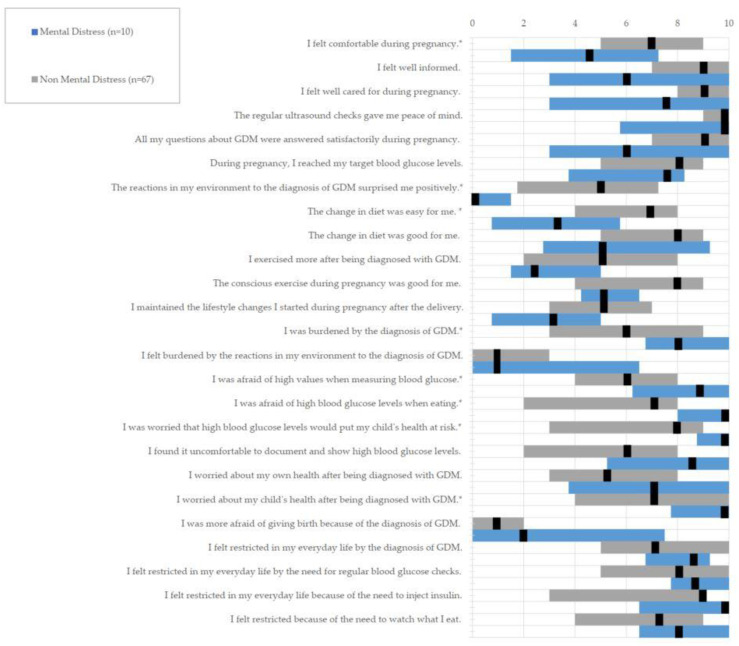
Association of mental distress and well-being during pregnancy obtained by the Psych-Diab-Questionnaire comparing the two subgroups ‘Mental Distress’ (n = 10) and ‘No Mental Distress’ (n = 67): scale from 0 to 10 with 0 being the lowest and 10 being the highest agreement to the question asked. The questions which showed significant differences are highlighted with *.

**Table 1 jcm-12-03358-t001:** Baseline characteristics of the total cohort and the subgroups with and without mental distress determined by SCL-R-90 questionnaire.

Variable	Total Cohort (n = 77)	No Mental Distress (n = 67; 87%)	Mental Distress (n = 10; 13%)	*p*
Age in years	34 (30–36)	33 (30–36)	36 (33.5–38.25)	0.092
Gravidity	1 (1.5–3)	2 (2–3)	2 (1–3)	0.159
Parity	1 (0–1)	1 (0–1)	1 (0–1.25)	0.437
Prepregnancy weight in kg	72.0 (61.5–87.5)	71 (60–84)	82 (71–91)	0.071
Prepregnancy BMI in kg/m^2^	26 (23–30)	25 (23–30)	29 (25–34)	**0.049 ***
Prepregnancy BMI categories				0.275
<18.5 kg/m^2^ (underweight)	1.3%	1.5%	-	
18.5–24.9 kg/m^2^ (normal)	44.2%	47.8%	20%	
25–29.9 kg/m^2^ (overweight)	27.3%	26.9%	30%	
≥30 kg/m^2^ (obesity)	27.3%	23.9%	50%	
History of GDM	20.8%	22.4%	10%	0.678
Thyroid disorders	26%	23.9%	40%	0.275
Cardiovascular disorders	3.9%	4.5%	-	1
Psychiatric disorders	5.2%	6%	-	1
Marital status				0.254
single	3.9%	4.5%	-	
married	44.2%	47.8%	2 (20%)	
permanent relationship	50.6%	46.3%	8 (80%)	
unknown	1.3%	1.5%	-	
Unemployment	6.5%	7.5%	-	0.467
Family History of Diabetes	63.6%	62.7%	7 (70%)	0.656
HbA1c at diagnosis in mmol/mol	32.2 (29.0–34.4)	31.15 (27.9–34.4)	32.2 (30.9–36.1)	0.171
75 g ogTT (in mmol/L)				
Fasting	5.3 (5.1–5.6)	5.3 (5.1–5.7)	5.1 (4.65–5.4)	0.074
1 h	9.7 (8.15–10.7)	9.6 (8–10.7)	10.1 (9.3–11)	0.365
2 h	7.5 (6.35–8.85)	7.45 (6.3–8.8)	8.1 (7.1–9.15)	0.224
SCL-90-R T-Scores				
Somatization (SOM)	54 (45–59)	51(43–59)	64 (59–74)	**<0.001 ***
Obsessive Compulsive (OC)	53 (43–60)	50 (43–56)	67 (63–71)	**<0.001 ***
Interpersonal Sensitivity (IS)	45 (42–55)	45 (39–51)	65 (62–70)	**<0.001 ***
Depression (DEP)	54 (44–60)	51 (43–58)	67 (64–72)	**<0.001 ***
Anxiety (ANX)	50 (39–59)	46 (39–56)	66 (61–80)	**<0.001 ***
Hostility (HOS)	52 (48–58)	52 (48–56)	63 (58–69)	**<0.001 ***
Phobic Anxiety (PHOB)	44 (43–56)	43 (43–53)	67 (59–74)	**<0.001 ***
Paranoid Ideation (PAR)	48 (40–54)	40 (40–51)	69 (63–70)	**<0.001 ***
Psychoticism (PSY)	43 (43–56)	43 (43–54)	65 (60–67)	**<0.001 ***
Global Severity Index (GSI)	52 (43–58)	50 (41–56)	66 (65–72)	**<0.001 ***
Positive Symptom Distress Index (PSDI)	58 (50–64)	55 (40–61)	65 (64–71)	**<0.001 ***
Positive Symptom Total (PST)	50 (42–56)	49 (42–53)	64 (59–67)	**<0.001 ***

Data are n (%) or median (interquartile range) unless otherwise specified. *p*—Comparison of subgroups with and without mental distress; * *p* < 0.05 is significant and bold. BMI—body mass index; GDM—gestational diabetes mellitus.

**Table 2 jcm-12-03358-t002:** Pregnancy characteristics and perinatal outcome of the total cohort (n = 77) and the subgroups with and without mental distress determined by SCL-R-90 questionnaire.

Variable	Total Cohort (n = 77)	No Mental Distress (n = 67; 87%)	Mental Distress (n = 10; 13%)	*p*
Hba1c at delivery in mmol/mol	34.4 (31.1–35.5)	33.33 (30.6–35.5)	34.95 (33.03–35.78)	0.305
Need for Insulin	41.6%	40.3%	50%	0.733
Treatment methods				0.14
Diet	58.4%	59.7%	50%	
Bolus	1.3%	1.5%	-	
Basal	29.9%	31.3%	20%	
Basal and bolus	10.4%	7.5%	30%	
Max. Insulin IU/kg	0.30 (0.19–0.4)	0.28 (0.18–0.4)	0.36 (0.2–0.44)	0.579
Max. number of injections	1 (1–2.75)	1 (1–2)	2 (1.5–4)	0.201
GWG in kg	12 (9–16.5)	12 (8.1–16.2)	11.3 (9.2–18.1)	0.939
Pregnancy complications	13.3%	15.2%	-	0.306
Pre-eclampsia/PIH/HELLP	1.8%	2%	-	1
IOL	41.9%	42.9%	33.3%	1
C-section	32.3%	33.9%	16.7%	0.654
Birth weight	3415 (3225–3672)	3402 (3226–3670)	3472 (3097–3705)	0.991
GA at delivery	39 (38–40)	39 (38–40)	38 (38–39)	**0.031 ***
SGA	1.6%	1.8%	-	1
LGA	9.8%	10.9%	-	1
5 min APGAR	9 (9–10)	9 (9–10)	9.5 (8.8–10)	0.454
pH	7.27 (7.2–7.32)	7.27 (7.20–7.32)	7.27 (7.20–7.32)	0.931
NICU admission	8.2%	7.3%	16.7%	0.415
hyperbilirubinemia	35.4%	36.4%	25%	1
hypoglycemia	14.8%	14.3%	20%	0.567
Respiratory distress	4.9%	5.5%	0	1

Data are n (%) or median (interquartile range) unless otherwise specified. *p*—Comparison of subgroups with and without mental distress; * *p* < 0.05 is significant and bold. GA—gestational age; GWG—gestational weight gain; IOL—induction of labor; SGA—small for gestational age; LGA—large for gestational age; NICU—neonatal intensive care unit; PIH—pregnancy induced hypertension; SGA—small for gestational age.

**Table 3 jcm-12-03358-t003:** Psych-Diab-Questionnaire Results of the total cohort and the two subgroups with and without mental distress determined by SCL-R-90 questionnaire.

Variable	Total Cohort (n = 77)	No Mental Distress (n = 67; 87%)	Mental Distress (n = 10; 13%)	*p*
I felt comfortable during pregnancy.	7 (4–9)	7 (5–9)	4.5 (1.5–7.3)	**0.030 ***
I felt well informed.	8 (7–10)	9 (7–10)	6 (3–10)	0.133
I felt well cared for during pregnancy.	9 (8–10)	9 (8–10)	7.5 (3–10)	0.130
The regular ultrasound checks gave me peace of mind.	10 (8.5–10)	10 (9–10)	10 (5.75–10)	0.458
All my questions about GDM were answered satisfactorily during pregnancy.	8 (5.5–9.5)	9 (7–10)	6 (3–10)	0.097
During pregnancy, I reached my target blood glucose levels.	8 (5–9)	8 (5–9)	7.5 (3.8–8.3)	0.501
The reactions in my environment to the diagnosis of GDM surprised me positively.	4.5 (0–7)	5 (1.8–7.3)	0 (0–1.5)	**0.003 ***
The change in diet was easy for me.	5 (4–8)	7 (4–8)	3.5 (0.8–5.8)	**0.037 ***
The change in diet was good for me.	8 (5–9)	8 (5–9)	5 (2.8–9.3)	0.190
I exercised more after being diagnosed with GDM.	5 (2–8)	5 (2–8)	2.5 (1.5–5)	0.093
The conscious exercise during pregnancy was good for me.	7 (4–9)	8 (4–9)	5 (4.3–6.5)	0.146
I maintained the lifestyle changes I started during pregnancy after the delivery.	5 (2–7)	5 (3–7)	3 (0.8–5)	0.054
I was burdened by the diagnosis of GDM.	7 (3.5–9)	6 (3–9)	8 (6.8–10)	**0.048 ***
I felt burdened by the reactions in my environment to the diagnosis of GDM.	1 (0–3)	1 (0–3)	1 (0–6.8)	0.708
I was afraid of high values when measuring blood glucose.	7 (4–8.5)	6 (4–8)	9 (6.3–10)	**0.022 ***
I was afraid of high blood glucose levels when eating.	8 (2–9)	7 (2–8)	10 (8–10)	**0.002 ***
I was worried that high blood glucose levels would put my child’s health at risk.	8 (4.5–10)	8 (3–9)	10 (8.8–10)	**0.005 ***
I found it uncomfortable to document and show high blood glucose levels.	6 (2–8)	6 (2–8)	8.5 (5.6–10)	0.051
I worried about my own health after being diagnosed with GDM.	6 (3.5–8)	5 (3–8)	7 (3.8–10)	0.313
I worried about my child’s health after being diagnosed with GDM.	8 (5–10)	7 (4–10)	10 (7.8–10)	**0.017 ***
I was more afraid of giving birth because of the diagnosis of GDM.	1 (0–2.5)	1 (0–2)	2 (0–7.5)	0.399
I felt restricted in my everyday life by the diagnosis of GDM.	8 (6.5–10)	7 (5–10)	8.5 (6.8–9.3)	0.370
I felt restricted in my everyday life by the need for regular blood glucose checks.	8 (6.5–10)	8 (5–10)	8.5 (7.8–10)	0.230
I felt restricted in my everyday life because of the need to inject insulin.	9 (4.5–10)	9 (3–9)	10 (6.5–10)	0.295
I felt restricted because of the need to watch what I eat.	7 (5–9)	7 (4–9)	8 (6.5–10)	0.229

Data are median (interquartile range) unless otherwise specified. *p*—Comparison of subgroups with and without mental distress; * *p* < 0.05 is significant and bold. GDM—gestational diabetes mellitus.

## Data Availability

The data presented in this study are available on reasonable request from the corresponding author.

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
