# Peer review of "Disease Perception and Mental Health in Pregnancies with Gestational Diabetes—PsychDiab Pilot Study"

_jcm, 2023, doi:10.3390/jcm12103358_

Round 1
Reviewer 1 Report
Esteemed Editor and Author Team,
This is an interesting research, worth publishing. However, some changes are required prior to acceptance:
1. Most of the introduction should be moved to Discussions since it deals with comparative observations regarding similar research.
2. It does not stem out from the Materials and Methods section when the questionnaires were applied, it seems it was done so after the 6 weeks postpartum visit. It should be clearly stated that this is a retrospective/historical study. To my knowledge, application of psychologic tests is not to be done for past events. None of the studies included in the references section is retrospective.
3. Results should be present either in the text, tables, or figures. Duplicates should be removed.
4. GDM patients monitoring and counseling falls in the hands of both the obstetrician and diabetician-nutritionist, this should also be stipulated by the study. Patient tailored medicine is supposed to be the standard of care today. Therefore, patients with history of mental illnesses or just mental distress should be carefully counseled by their attending physicians. A psychologist or psychiatrist is a mandatory third member of the team involved in patient care.
5. GDM diagnosis is different for low and high risk patients. The authors do not state which kind of patients were included in the study, just the WHO diagnosis criteria. The use of HbA1c for GDM diagnosis is still a matter of controversy.
6. It is interesting and should be detailed the large percentage of NICU admissions (16,7%) for term deliveries (38-39 weeks of gestation) in mentally distressed patients. The same parameter is 7.3% in control patients who delivered at 39-40 weeks.
7. Were the patients given corticoprophylaxis at any time?
8. The study does not mention if there were any cases of neonates with respiratory distress (there was a large number of GDM patients who required insulin therapy).
Reviewer 2 Report
I would suggest reformulating the first sentence of the Results, because the authors have already mentioned in the methodology the number of patients included and the time interval, this is information that is repeated.
In figure 2 and half of table 1, the same information is presented twice, regarding the results SCL-90-R questionnaire and the comparison of the Mental Distress and 'No Mental Distress' groups. I do not see the difference between the information shown in table 2 and figure 3.
Phrases from Discussions such as: When looking at the findings of our Psych-Diab-Questionnaire (see Figure 2 and Table 3), there is a clear trend that patients with mental distress answered positive questions with lower mean scores (7.5 vs. 5) and negative questions with higher mean scores (6 vs. 7.7)…. should be avoided; The groups are not balanced (10 patients vs 67 controls) and many associations are not statistically significant, therefore should be rewritten as such.
The conclusions should be strictly based on the results of the study. The analogy with postpartum depression was not the subject of the presented study.
Reviewer 3 Report
The study "Disease perception and mental health in pregnancies with gestational diabetes - pyschDiab pilot study" is an excellent study. Mental health among our pregnant and postpartum patients with diabetes is an area of much needed research, which is highlighted by this manuscript. The study may be strengthened by comparing baseline characteristics (that may be available by chart review/abstraction) of those ~180 subjects that did not respond. For example, did the subjects who did not respond have similar age, ethnicity, employment status, insulin use, neonatal outcomes, pregnancy complications, BMI, etc, compared to the 30% of the 257 patients that responded. Overall, this was a very well-written, organized summary of an interesting pilot study.
Round 2
Reviewer 2 Report
The authors adequately addressed the issues raised